# Trans-Kingdom RNA Dialogues: miRNA and milRNA Networks as Biotechnological Tools for Sustainable Crop Defense and Pathogen Control

**DOI:** 10.3390/plants14081250

**Published:** 2025-04-20

**Authors:** Hui Jia, Pan Li, Minye Li, Ning Liu, Jingao Dong, Qing Qu, Zhiyan Cao

**Affiliations:** 1State Key Laboratory of North China Crop Improvement and Regulation, Hebei Agricultural University, Baoding 071000, China; jiahui@hebau.edu.cn (H.J.); lipan@hebau.edu.cn (P.L.); lning121@126.com (N.L.); shmdig@hebau.edu.cn (J.D.); 2College of Life Sciences, Fujian Normal University, Fuzhou 350117, China; 108032024026@student.fjnu.edu.cn; 3College of Agriculture and Forestry, Hebei North University, Zhangjiakou 075000, China

**Keywords:** plant miRNA, fungal milRNA, plant–pathogen interaction, endogenous regulation, trans-kingdom regulation

## Abstract

MicroRNAs (miRNAs) are a class of non-coding RNAs approximately 20–24 nucleotides in length, which play a crucial role during gene regulation in plant–pathogen interaction. They negatively regulate the expression of target genes, primarily at the transcriptional or post-transcriptional level, through complementary base pairing with target gene sequences. Recent studies reveal that during pathogen infection, miRNAs produced by plants and miRNA-like RNAs (milRNAs) produced by fungi can regulate the expression of endogenous genes in their respective organisms and undergo trans-kingdom transfer. They can thereby negatively regulate the expression of target genes in recipient cells. These findings provide novel perspectives for deepening our understanding of the regulatory mechanisms underlying plant–pathogen interactions. Here, we summarize and discuss the roles of miRNAs and milRNAs in mediating plant–pathogen interactions via multiple pathways, providing new insights into the functions of these RNAs and their modes of action. Collectively, these insights lay a theoretical foundation for the targeted management of crop diseases.

## 1. Introduction

Fungi, bacteria, and viruses are the primary pathogens responsible for plant diseases. Different pathogens employ distinct pathogenic mechanisms to damage a host. Fungi produce cellulase, pectinase, protease, and other enzymes that degrade the cellulose, pectin, and protein components of the plant cell wall, thereby compromising the plant’s structural integrity and weakening its first line of defense [1]. Upon invading plant tissues, necrotrophic fungi secrete toxins that induce cytoplasmic lysis and release cytochromes, which disrupt the integrity of the plant cell membranes and impair cellular morphology and function, ultimately resulting in necrosis and tissue death [2]. Plants are exposed to various pathogens that can significantly impact growth, development, and yield. To defend against these threats, plants have developed two layers of immune responses: pathogen-associated molecular pattern-triggered immunity (pattern-triggered immunity, PTI) and effector-triggered immunity (effector-triggered immunity, ETI) [3,4].

During the initial stages of infection, PTI provides the first line of plant defense; it is activated when pathogen-associated molecular patterns (PAMPs) from pathogens are detected via interaction with pattern recognition receptors (PRRs), which are located on the plant cell surface, either on the plasma membrane or cell wall. This ligand-receptor interaction initiates the perception of pathogen-derived invasion signals. Subsequently, PRR-mediated recognition triggers intracellular signaling cascades, including a rapid influx of Ca^2+^ and a reactive oxygen species (ROS) burst, which together establish localized immune responses to limit pathogen progression. PTI causes a range of defense responses in plants.

However, PTI is usually less efficient than ETI, which is gene-for-gene specific, a more robust defense mechanism mediated by direct or indirect recognition of pathogen effectors.. ETI is the second layer of defense triggered by pathogen effectors and interacts with the R proteins; it often causes robust and rapid defensive responses, such as hypersensitive response (HR) and necrotic cell death [5,6].

Systemic acquired resistance (SAR) is a long-lasting and broad-spectrum resistance incurred in the cells distant from the local infections, which are triggered by either PTI or ETI and transferred using salicylic acid (SA) as signals [7]. SA is a key signaling molecule for the induction of SAR. When a plant is locally attacked by a pathogen, the SA level at the attacked site rapidly increases, and then the SA signal is transmitted to other parts of the plant, triggering a series of defense responses and enabling the plant to acquire broad-spectrum resistance to a wide range of pathogens.

The involvement of plant hormonal regulators is crucial for mediating complex defense pathways against biotic stresses. Plant hormones, including SA, jasmonic acid (JA), and ethylene (ET), are vital signaling molecules in plants. The hormone signaling pathways are rapidly activated and involved in various aspects of the immune response. SA acts as the central signaling molecule in plant immunity against biotrophic and hemibiotrophic pathogens, primarily through the activation of SAR [8,9]. In contrast, JA and ET primarily mediate defense responses to necrotrophic pathogen invasion and insect herbivory through induced systemic resistance pathways [10,11,12]. During the initial biotrophic phase of hemibiotrophic infection, plants can activate SA-mediated responses to limit pathogen colonization and suppress the establishment of the biotrophic interface. However, as the pathogen transitions to the necrotrophic phase, JA signaling becomes increasingly important for counteracting tissue damage and combating the necrotrophic attack. Simultaneously, complex signaling interactions occur among various hormones, forming a precise regulatory network that finely tunes the immune responses of plants during different pathogen infections [13,14].

MicroRNA (miRNA) is a class of endogenous, non-coding small RNAs that regulate the expression of target genes at the transcriptional level. Their origin can be traced back to early eukaryotes. miRNAs are derived from intergenic regions [15], intronic regions [16,17], and repetitive elements [16] in the genome. In 1993, Victor Ambros’ team [18] identified the first miRNA in *Caenorhabditis elegans*: lin-4, which regulates larval development by inhibiting expression by binding to the 3′-untranslated region (3′ UTR) of lin-14 mRNA. In 2000, Reinhart et al. [19] discovered let-7, which also plays a key role in nematode development. Subsequently, Reinhart et al. [20] detected the presence of miRNAs in *Arabidopsis thaliana* by Northern blot, suggesting that miRNAs also play an important role in plants. miRNA sequences are highly conserved in evolution; for example, let-7 in nematodes has homologous sequences in Drosophila, zebrafish, and even humans [21], which suggests that its function is widely conserved in evolution.

In the nucleus of plant cells, RNA polymerase II transcription produces the primary transcript pri-miRNA with a cap structure at the 5′ end and a polyadenylate tail (polyA) at the 3′ end. The zinc finger protein SERRATE (SE), Cap-Binding Complex (CBC), dsRNA binding protein (HYPONASTIC LEAVES 1, HYL1), and Dicer-like protein (DCL) co-localize in the nucleus to form 2–4 cleavage vesicles (dicing body) with a diameter of 0.2–0.8 μm [22]. The DCL proteins belong to the ribonuclease III (RNase III) family and have a conserved helicase domain, DUF283 domain, PAZ domain, two RNase III domains, and a C-terminal dsRBD. DUF283 can link deconjugase and PAZ; the PAZ structural domain has a 3′ binding pocket that connects the protruding bases at the 3′ end of RNA by hydrogen bonding, and the main function of the dsRBD is to bind to dsRNA. The 2 RNase III domains of the DCL protein, RNase IIIa, and RNase IIIb, form a dimer, which is the cleavage site of dsRNA. In the presence of Mg^2+^, the RNase IIIa structural domain cleaves the 3′ hydroxyl-containing RNA strand, and the RNase IIIb structural domain cleaves the 5′ phosphoric acid-containing RNA strand, catalyzing the hydrolysis of the phosphodiester bonds within each strand of dsRNA to produce a 20–24 bp RNA fragment. Using the above domains, DCL cleaves pri-miRNA twice to form a double-stranded miRNA/miRNA* complex [23]. How does DCL recognize the pre-miRNA of miRNA: miRNA* and thus cleave it? Miskiewicz et al. [24,25] used bioinformatics tools such as WebLOGO, MEME Suite, RNA structure, and mfold to analyze 50 sequences with experimentally confirmed miRNA/miRNA* cleavage mechanisms and the predicted structures of 5975 sequences. The results showed that pre-miRNAs have symmetric internal loops 1-1 (single unpaired nucleotide on each strand of the vicinity region) and 2-2 internal loops (two unpaired nucleotides on each strand of the vicinity region). MiRNA double-stranded vicinity symmetric internal loops (1-1 and 2-2) are key signals recognized by DCL and can direct cleavage by destabilizing the double strand.

The miRNA/miRNA* complex formed by DCL cleavage can be detected by HUAENHANCER1 (HEN1), which modifies the last base at the 3′ end of each strand in the complex by methylation, making its structure more stable and less susceptible to degradation [26]. With the formation of the miRNA/miRNA* complex, the number of pri-miRNAs in the cleavage vesicle decreases, the force between HYL1 and SE proteins is weakened, and the complex carrying miRNA/miRNA* migrates outwards. It is released to the outside of the cleavage vesicle, and the complex is translocated to the cytoplasm by the HASTY proteins. The double-stranded miRNA/miRNA* complex is separated by RNA helicases. One strand is degraded, and the other is a mature miRNA. The mature miRNA can bind to the Argonaute protein (AGO) and form the miRNA-inducing silencing complex (miRISC) [22]. Mature miRNA primarily recognizes target genes through complementary base pairing. It cleaves the target mRNA when there is perfect complementarity with the target gene and inhibits translation when complementarity is imperfect (Figure 1) [27,28].

The presence of miRNAs has been demonstrated in plants. Although initially thought to be absent in fungi, miRNA analogs were later discovered in *Neurospora crassa*, referred to as miRNA-like (milRNA) due to their similarity to plant miRNAs [29]. Heng-Chi Lee et al. [29] provided a comprehensive review of milRNA synthesis, which differs from the plant miRNA synthesis pathway by involving four distinct modes, each requiring a different set of proteins. milR-1 is a small RNA molecule with lengths of 19 nt, 24 nt, and 25 nt, which is transcriptionally synthesized by RNA polymerase III. In the presence of DCL proteins, the pri-milRNA precursor with a stem-loop structure is first cleaved into pre-milRNA, which then binds to QUELLING DEFICIENT-2 (QDE-2, an Argonaute protein), which, in turn, recruits exonuclease QIP proteins with nucleic acid exonuclease activity to be processed into mature milRNAs, where QDE-2 provides the processing platform for mature miRNA synthesis. The synthesis of milR-2 is completely independent of DCL proteins. It relies instead on QDE-2 proteins. MRPL3 possesses a putative RNAse III domain and a dsRNA recognition motif and is responsible for the processing of the pri-milRNA of milR-2 into pre-milRNAs. Subsequently, QDE-2 binds to pre-milRNAs, and its slicer activity plays a role in the production of mature milRNA. milR3 is synthesized in a manner similar to miRNA synthesis in plants, relying entirely on DCL proteins. The synthesis of milR-4 is partially dependent on DCL proteins, and unlike milR-1, in which QDE-2 does not bind to pre-miRNAs but to mature miRNAs, the synthesis of milR-4 requires the involvement of MRPL3 (Figure 2). Although research on fungal milRNAs began relatively late, the development of high-throughput sequencing technology has led to the discovery of an increasing number of fungal milRNAs. Notably, 49 milRNAs were detected in *Fusarium graminearum* [30]. In *Botryosphaeria dothidea*, 167 conserved *Bd*-milRNAs and 68 novel *Bd*-milRNAs were identified [31]. Thirteen novel and predicted milRNAs were detected in *Trichoderma reesei* [32]. Nine milRNAs were confirmed in *Verticillium nonalfalfae* [33]. Ten milRNAs were detected in *Fusarium oxysporum* through RT-PCR [34]. A total of 51 novel milRNAs were identified in *Fusarium verticillioides* [35]. Two milRNAs and 42 milRNA candidates were verified in *Sclerotinia sclerotiorum* [36].

MiRNAs are involved in various aspects of plant growth, development, and responses to salt and drought stress [37,38,39,40]. Recent studies have shown that miRNAs (milRNAs) play a crucial role in plant–pathogen interactions. MiRNAs (milRNAs) are transferred bidirectionally between host and pathogen, crossing species boundaries and translocating across species to target specific genes [41,42]. They regulate the expression of receptor genes in both host and fungal cells and are involved in the host’s disease resistance and the pathogenesis of fungi. Bidirectional delivery of miRNAs (milRNAs) has been observed in animal, plant, and fungal interactions, and it influences host–fungal interactions [43,44]. This review summarizes the dual regulatory networks of plant miRNAs and fungal milRNAs in plant–pathogen interactions. On the one hand, plant miRNAs are involved in PTI, ETI, and hormone signaling pathways through endogenous gene regulation mechanisms. In contrast, fungal milRNAs are downregulated or not expressed. They negatively regulate the expression of endogenous genes related to pathogenicity or virulence, which are involved in the pathogenicity process. On the other hand, both miRNAs and milRNAs exhibit complex mechanisms of trans-kingdom regulation. Plant miRNAs are delivered to fungi via exosomes or extracellular vesicles, where they inhibit the expression of virulence factor mRNAs by binding to them through sequence complementarity. In contrast, fungal milRNAs can be transferred to plants, inducing gene silencing through the plant’s endogenous RNA-mediated interference (RNAi) pathway. This reduces the plant’s ability to resist disease and creates favorable conditions for pathogen infection. This review highlights the dual role of miRNAs as “molecular signals” and “defense weapons” in plant–pathogen interactions, and it lays the foundation for miRNA studies and offers insights for disease prevention and control.

## 2. miRNA (milRNA) Regulates Endogenous Gene Expression

### 2.1. miRNAs Regulate Endogenous Gene Expression and Participate in Disease Resistance

Over the course of the long-term co-evolution between plants and pathogens, two key immune defense mechanisms, PTI and ETI, have gradually evolved. Increasing evidence suggests that miRNAs play a crucial role in these processes [45,46,47,48,49,50,51] (Figure 3A).

#### 2.1.1. miRNAs Regulate PTI

During PTI, the plant recognizes PAMPs through PRRs, initiating a series of defense responses. miRNAs are involved in various signaling pathways. They play a crucial regulatory role in the initiation and modulation of PTI. miR393 is the first miRNA identified as being closely associated with plant resistance. miR393 was upregulated when *Arabidopsis* sensed the bacterial flagellin protein flg22, inhibiting receptor expression by targeting the mRNA of F-box auxin receptor genes TIR1, AFB2, and AFB3. Inhibition of the auxin signaling pathway limits the growth of *Pseudomonas syringae*. miR393 activates the PTI response by inhibiting the growth hormone signaling pathway, ultimately enhancing *Arabidopsis* resistance to *P. syringae* pv. tomato DC3000 [52]. This process clearly illustrates the molecular mechanism by which miRNAs regulate PTI. By regulating key signaling pathways, miRNAs enhance plant disease resistance.

After recognizing PAMPs through PRRs, plant cells activate various signaling pathways in PTI responses. PRR proteins are involved in the perception of pathogen invasion signals and early signal transduction pathways. Upon recognition of pathogen signals, PRRs rapidly activate associated enzymes, such as NADPH oxidase, resulting in the rapid intracellular production and accumulation of ROS. ROS mainly include superoxide anion (O^2−^), hydrogen peroxide (H_2_O_2_), and hydroxyl radicals (OH^+^). They regulate physiological and metabolic processes in plant cells through redox signaling in response to pathogen invasion, thereby enhancing plant resistance to pathogens. The ROS burst activates downstream signaling pathways, such as the mitogen-activated protein kinase (MAPK) cascade, which in turn induces the expression of genes involved in callose synthesis, leading to callose deposition in the cell wall [5]. This plays a critical role in plant disease resistance. The expression of miR-400 in *Arabidopsis thaliana* was downregulated when the plant was infected with *B. cinerea*. Furthermore, the expression of its endogenous target genes, which encode pentatricopeptide repeat (PPR) proteins, was upregulated. Plants overexpressing miR-400 and *ppr*-deficient mutants were more sensitive to *B. cinerea*. miR-400 responded to pathogen infection by regulating the expression of PPR mRNAs [53]. miR-398 responded to pathogen infection by promoting plant H_2_O_2_ production through a series of superoxide dismutase enzymes [54,55,56,57]. In *Magnaporthe oryzae*-infected rice, miR-1871 was downregulated to inhibit the expression of endogenous target genes encoding a microfibrillar-associated protein gene (*MFAP1*) in miR-1871-blocking lines. This induced increased ROS production and callus accumulation in plants inhibited pathogen infection, and increased rice yield [46]. In addition, *Arabidopsis* can interact with necrotrophic (*Plectosphaerrella cucumerina*) and hemibiotrophic (*F. oxysporum*, *Colletotrichum higginianum*) fungi. *MIM773* is a transgenic line for the miR773 target mimic. *MIM773* plants exhibited higher callose and ROS accumulation than wild-type plants during inoculation with *P. cucumerina*. *MIM773* plants exhibited stronger PTI upon *P. cucumerina* infection [58].

#### 2.1.2. miRNAs Regulate ETI

Nucleotide-binding site leucine-rich repeat (*NBS-LRR*) genes play a crucial role in the ETI response. The proteins encoded by these genes contain distinct NBS and LRR structural domains [59,60]. miRNAs precisely regulate *NBS-LRR* gene expression by targeting conserved domains within these genes. For instance, under phytopathogenic stress conditions, such as infection with *Verticillium dahliae*, potato miR-482e was downregulated. It mediated the transcriptional repression of its cognate *NBS-LRR* target genes. *NBS-LRR* functions as a PHAS (Phased, Secondary siRNAs) locus, triggering the biogenesis of trans-acting small interfering RNAs (ta-siRNAs). These orchestrate signal transduction amplification cascades, thereby fine-tuning plant defense mechanisms [61]. Typically, after being guided and cleaved by miRNAs or other siRNAs, the PHAS loci are processed by enzymes such as RNA-dependent RNA polymerase 6 (RDR6) and Dicer-like 4 (DCL4) to generate phased 21 nt or 24 nt phasiRNAs [62]. These siRNAs can further regulate the expression of other genes through perfect complementary base pairing. Furthermore, host miR-482 contributes to plant disease resistance by regulating the expression of *NBS-LRR* during interactions between *V. dahliae* and cotton, *F. oxysporum* and tomato, and *Colletotrichum gloeosporioides* and poplar [63,64,65]. These studies provide further compelling evidence for the role of miRNAs in the ETI response through the regulation of *NBS-LRR* genes in disease resistance.

#### 2.1.3. miRNAs Regulate Hormones Signal Transduction

Phytohormones act as master regulators, orchestrating plant adaptive responses to both biotic and abiotic stresses. These signaling molecules operate within intricately interconnected networks, mediating stimulus-specific transcriptional reprogramming via coordinated receptor-ligand interactions and downstream signal transduction cascades [8,66,67]. Overexpression of miR-160 in *Arabidopsis* results in enhanced callose accumulation and increased resistance to pathogens. miR-160 has been shown to regulate the expression of growth hormone response factors *ARF-10*, *ARF-16*, and *ARF-17* [68]. When plants are infected with a pathogen, SA accumulates in the plant and is sensed by its receptor *NPR1* (Non-expresser of *PR* genes 1), which activates the expression of downstream transcription factors *TGA* and *WRKY* [69]. In the tomato, *Sl*-miR396a negatively regulates the transcription of *GRF1* and reduces the expression levels of *TGA1/2* and *PRs*. Lines overexpressing *Sl*-miR396a were generated and exhibited increased disease susceptibility, suggesting that *Sl*-miR396a regulates plant disease resistance through the SA pathway [52]. miR477 is upregulated and negatively regulates its target gene *GhCBP60A* during *V. dahliae* infection in cotton. In the SA signaling pathway, *GhCBP60A* has been shown to negatively regulate the expression of *ICS1* and *PR1* genes. In *GhCBP60A*-deficient plants, the SA level was significantly increased. This, in turn, activates a series of immune responses involved in plant resistance to the pathogen [70]. This suggests that miR477 may indirectly affect the SA pathway through the regulation of *GhCBP60A* and play a critical role in plant defense mechanisms against pathogen infection [71]. A novel maize miRNA, zma-unmiR4, regulates the expression of the endogenous gibberellin-related gene *ZmGA2ox4*, affecting maize growth and resistance to *F. verticillioides* [72].

### 2.2. Fungal milRNAs Regulate Endogenous Gene Expression and Are Involved in Pathogenesis

Pathogenic fungi participate in plant–pathogen interactions by secreting effector proteins and metabolites. Recent evidence suggests that fungal milRNAs also play a role in pathogenic processes by regulating the expression of endogenous genes. The endogenous target genes of fungal milRNAs are related to pathogenicity or virulence. MilRNAs negatively regulate the expression of target genes by binding to their mRNA sequences and cleaving the mRNAs through base complementary pairing (Figure 3B). The abundance of fungal milRNAs is normal when the plant is uninfected. When the plant is infected, the abundance of milRNAs is downregulated or is not detected. This leads to the upregulation of target genes involved in disease progression, successful infection, and colonization of the plant. For example, *Vm*-milR37 from *Valsa mali* can be detected in the fungal mycelium but was not detected during apple infection. Strains overexpressing *Vm*-milR37 did not affect the nutritional growth of the fungus but reduced its pathogenicity [73]. Upon pathogen invasion, the abundance of *Vm*-milR16 is decreased, releasing repression of its endogenous target genes—secreted protein (*VmSP1*), sucrose non-fermenting 1 (*VmSNF1*), 4,5-DOPA dioxygenase estradiol (*VmDODA*), and a hypothetical protein (VmHy1)—which are upregulated and involved in the pathogenic process [74,75]. The abundances of 51 milRNAs were detected in the hyphae of *F. verticillioides*, but the abundances of milRNAs were not detected in the interaction samples. This suggests that upon pathogen infection, the abundances of milRNAs in the fungus either decreased or became completely absent. Endogenous pathogenic or virulence-associated target genes may be upregulated to participate in pathogenicity. The specific mechanisms of action require further investigation to clarify the function of these genes [35].

In contrast, the abundance of *Foc*-milR-87 is increased during the interaction between *F. oxysporum* f. sp. *cubense* (Foc) and banana. *Foc*-milR-87 specifically targets the endogenous virulence gene *FOIG_15013*, which encodes a glycosyl hydrolase. *FOIG_15013* plays a key role in the interaction between banana and *Foc*. It induces the expression of disease-resistance genes in bananas, including disease-related protein 1 (*PR-1*), phenylalanine ammonia-lyase 4 (*PAL4*), lipoxygenase (*LOX*), and osmoregulatory protein (*Osmotin*). The *Foc*-milR-87 knockout mutant and overexpressing strains were constructed and evaluated for pathogenicity. The results showed that the pathogenicity of the *Foc*-milR-87 knockout mutant was significantly reduced, while that of the overexpressing strain was significantly enhanced. Notably, the pathogenicity of the *FOIG_15013* deletion mutant was similar to that of the *Foc*-milR-87 overexpressing strain. The results of this series of experiments strongly suggest that during the interaction between *Foc* and banana, the milRNAs produced by the fungus (e.g., *Foc*-milR-87) may reduce banana disease resistance by increasing their abundance, which negatively regulates the target genes (e.g., *FOIG_15013*) and increases the pathogenicity of the fungus [76].

## 3. miRNAs (milRNAs) Mediate Trans-Kingdom Regulation in Plant–Pathogen Interactions

The involvement of miRNAs and milRNAs in trans-kingdom interactions between plants and pathogens has been increasingly studied. These small non-coding RNAs are produced by both host plants and pathogens. They play regulatory roles in modulating gene expression across species boundaries [77,78,79] (Table 1). For example, plant miRNAs may target pathogen virulence genes, while pathogen milRNAs may suppress host immune responses [35,74,80]. This bidirectional RNA-mediated communication reveals a sophisticated molecular interplay in the co-evolution of plants and pathogens. Current research focuses on elucidating the mechanisms of RNA trafficking, stability, and target specificity in these trans-kingdom regulatory networks.

### 3.1. Plant miRNAs Modulate Fungal Gene Expression and Suppress Pathogen Invasion

Over the course of the long evolutionary process of plant–pathogen interactions, plants have developed a series of complex and subtle defense mechanisms against fungal attacks. The success of host-induced gene silencing (HIGS) has introduced a novel approach for the prevention and control of plant fungal diseases. HIGS effectively protects plants from fungal infections by silencing pathogen genes [81,82,83]. However, as research advanced, a critical question arose: Can endogenous small RNAs (sRNAs), particularly miRNAs, be actively transferred across kingdoms into fungi, thereby regulating fungal gene expression? This question has become a prominent research topic in the field of plant–pathogen interactions.

Recent studies have demonstrated that plant miRNAs can be translocated to fungi via extracellular vesicles (EVs) during pathogen infection [42,84] (Figure 4). EVs are membranous spherical vesicles produced and secreted by prokaryotic and eukaryotic cells, and they play a crucial role in the molecular exchange between plants and pathogenic microorganisms [85]. EVs secreted by eukaryotic cells include apoptotic bodies, microvesicles, and exosomes. Exosomes are a type of extracellular vesicle secreted by cells, typically with a diameter ranging from 40 to 200 nm. They are released after the fusion of multivesicular bodies (MVBs) with the cell membrane and play important roles in intercellular communication and the regulation of physiological and pathological processes [86]. The translocation of *Malus hupehensis* miR159a into the fungal protoplast occurred via exosomes [87]. These miRNAs transported to fungi can reduce fungal pathogenicity by targeting and downregulating genes involved in fungal virulence and pathogenicity [42,84]. RNA-binding proteins (RBPs) are involved in plant immune responses by selecting and stabilizing sRNAs targeting fungal genes to facilitate their transfer. For example, Baoye He et al. [88] showed that RBPs such as AGO1, RNA helicase 11 (RH11), and RH37 specifically bind to sRNAs such as miR166, TAS1c-siR483, and TAS2-siR453 that are enriched in EVs, whereas annexin 1 (ANN1) and ANN2 bind non-specifically and may play a role in stabilizing sRNAs. The RNA-binding protein hnRNPA2B1 mediates the selective sorting of miRNAs into EVs by recognizing specific sequence motifs (e.g., GGAG motifs [89], UAG motifs [90]) of miRNAs. The above findings uncover a novel mechanism of plant defense, suggesting that plants are not merely passive responders to pathogen attack but can also proactively inhibit pathogen infection through gene expression regulation through trans-kingdom regulation (Figure 3C). This speculation offers significant insights into the study of the trans-kingdom transfer of plant miRNAs. However, it is not yet clear how the miRNAs in the EVs get into the fungal cells, and further studies are needed to clarify the mechanism of their transfer. The above stimulates further exploration of their mechanisms of action.

In our study, we identified the important role of maize miR528b-5p in the interaction with *F. verticillioides*. Maize miR528b-5p is transferred to *F. verticillioides* during the interaction. Further studies revealed that miR528b-5p repressed the expression of *FvTTP*, which encodes a dual-membrane-spanning protein in *F. verticillioides*. We observed that the *FvTTP* deletion mutant did not differ significantly from the wild type in nutrient growth, but its pathogenicity and virulence were reduced, demonstrating that miR528b-5p critically influences the pathogen’s infection process [35]. However, the specific mechanism underlying the transfer of miR528b-5p remains unclear and requires further investigation.

### 3.2. Fungal milRNAs Reduce Plant Immunity by Regulating Disease-Resistance Gene Expression

Recent studies have demonstrated that fungal milRNAs can function as novel effectors in plant–pathogen interactions. MilRNAs can be transferred into host cells, where they suppress plant immunity by hijacking the host RNAi pathway [91] (Figure 3D). The discovery of the trans-kingdom transfer of fungal milRNAs into plants has provided new insights into the mechanisms underlying plant–pathogen interactions. Fungal milRNAs have been detected in plants through the sequencing and analysis of small RNAs from fungal–plant interaction samples. For example, *B. cinerea*-derived siRNAs were detected in *Arabidopsis thaliana* samples infected with *B. cinerea* using high-throughput sequencing. These siRNAs inhibit the expression of *Arabidopsis MPK2*, *MPK1*, *PRXIIF*, and *WAK* genes by hijacking the host AGO1. Knocking down *AtAGO1* abolished the regulatory effect of siRNAs on target gene expression [91]. A similar phenomenon was subsequently observed in the study of the interaction between *F. oxysporum* and tomato. *F. oxysporum*-derived *Fol*-milR1 was detected in tomato samples infected with *F. oxysporum*. It also binds to the tomato SlyAGO4a protein, hijacking the host RNAi pathway and thereby suppressing host immunity [92]. In addition to regulating the expression of endogenous virulence genes, as mentioned above [76], it can be transferred into banana to target the *MaPTI6L* gene (a pathogenesis-related gene encoding a transcriptional activator). *MaPTI6L* binds to the GCC box in the promoter of *MaEDS1*. This activation leads to the expression of downstream resistance-associated SA pathway genes. *MaPTI6L* is involved in the disease-resistance process [93]. *Foc*-milR138 specifically targets and inhibits the expression of the plant surface receptor-like kinase (*MaLYK3*) mRNA after translocation to banana during pathogen interaction. This molecular interference directly disrupts the *MaLYK3*-mediated immune signaling pathway. Such disruption leads to marked inhibition of resistance gene expression. Concurrently, it suppresses both ROS accumulation and localized callose deposition at infection sites. These coordinated alterations ultimately attenuate the plant’s disease resistance, thereby promoting *Foc* colonization and successful infection pathogenesis [94]. This study further enriches research on the trans-kingdom regulation of plant–pathogen interactions by fungal milRNA.

In our study, 51 *F. verticillioides* milRNAs were successfully identified. These milRNAs were found to regulate 333 genes in maize, as determined by bioinformatics analysis and experimental validation. GO and KEGG enrichment analyses revealed that these target genes are associated with biological pathways, including MAPK signaling, plant hormone signaling, and plant–pathogen signaling. However, the specific regulatory mechanisms through which *F. verticillioides* milRNAs affect these pathways remain unclear. Further in-depth studies are required to comprehensively analyze their molecular regulatory networks and provide a theoretical foundation for the prevention and control of maize diseases. Increasing evidence suggests that the trans-kingdom regulation of fungal milRNAs is a widespread phenomenon. The presence of fungal milRNAs in plants has also been demonstrated in interactions between *V. mali*, *Puccinia striiformis* f. sp. *tritici*, and their host plants. These milRNAs have been shown to regulate the expression of disease-resistance genes in host plants [41,74]. These studies not only further confirm the transfer of fungal milRNAs but also elucidate the molecular mechanisms by which they regulate plant disease resistance in plant–pathogen interactions. This provides a new perspective for a more comprehensive understanding of plant–pathogen interactions.

Fungal milRNAs can be transferred to plants in order to participate in the interaction, so what is the mechanism of the transport? Fungi secrete sRNAs when infecting plants, and sRNAs are transported outside of plant cells by EVs. In plants, clathrin-mediated endocytosis (CME) is the main pathway for cellular uptake of extracellular substances. Plant cells take up EVs containing fungal sRNAs via the CME pathway, which in turn allows fungal sRNAs to enter the plant cell interior, playing a role in regulating plant target genes by hijacking host AGO1 proteins [91]. *B. cinerea* uses EVs to secrete Bc-sRNAs, which are subsequently internalized by plant cells via CME [95] (Figure 4). A large number of *Arabidopsis* clathrin-coated vesicles (CCVs) surround the site of *B. cinerea* infection. The EVs marker gene of *B. cinerea*, tetratransmembrane protein Punchless 1 (*BcPLS1*), co-localizes with one of the core components of *Arabidopsis* CCVs, clathrin light chain 1 (*CLATHRIN LIGHT CHAIN 1*). At the same time, *BcPLS1* and sRNAs secreted by *B. cinerea* were detected in post-infection purified CCVs. The amount of *Bc*-sRNAs was reduced by 60% and 80% in knockout mutants of key *Arabidopsis* CME pathway genes, *chc2-1* and *ap2σ*, respectively. The above studies indicate that fungal milRNAs are encapsulated by EVs during plant–pathogen interactions, preventing the degradation of milRNAs and transferring them to plant cells for action through the CME pathway. The above findings provide important information for understanding plant–pathogen interactions, help to elucidate plant immune mechanisms, and provide a theoretical basis for developing novel control strategies against fungal diseases, such as considering the regulation of plant CME pathway-related genes for enhancing plant resistance to fungi.

### 3.3. Trans-Kingdom Regulation of miRNAs in the Mycorrhizal Symbiosis

In addition to negatively regulating target genes in pathogen interactions, miRNAs can also benefit both fungi and their hosts through trans-kingdom regulation. Studies on rhizobial nodulation and *arbuscular mycorrhizal* colonization have shown that beneficial microorganisms can employ miRNA-mediated signaling to promote symbiogenesis [96,97].

The colonization of *Eucalyptus grandis* roots by *Pisolithus microcarpus* facilitates the trans-kingdom transfer of fungal milRNAs. It has been shown that *Pmic*_miR-8 dynamically regulates the expression of the NB-ARC domain in root tissues, thereby maintaining fungal infection. Crucially, this regulated infection facilitates enhanced soil nutrient uptake by the host plant. This RNA-mediated molecular dialogue exemplifies a sophisticated mechanism underlying plant-fungal mutualism, highlighting its ecological and agronomic significance [98].

**Table 1 plants-14-01250-t001:** sRNA trans-kingdom regulation in plant–pathogen interaction.

Host Plant	Pathogen	sRNAs	Target Genes	Directionality of RNA Mobility	Reference
*Zea mays*	*Fusarium verticillioides*	miR528b-5p	*FvTTP*	Host-pathogen	[35]
cotton	*Verticillium dahliae*	miR166, miR159	Clp-1, HiC-15	Host-pathogen	[42]
*Malus hupehensis*	*Botryosphaeria dothidea*	miR159a	BdSTP	Host-pathogen	[87]
*Arabidopsis thaliana*	*Botrytis cinerea*	Bc-sRNAs	*MPK2*, *MPK1*, *PRXIIF*, and *WAK*	Pathogen-host	[91,95]
tomato	*Fusarium oxysporum*	*Fol*-milR1	SlyAGO4a	Pathogen-host	[92]
banana	*Fusarium oxysporum*	Foc-milR87	*MaPTI6L*	Pathogen-host	[93]
banana	*Fusarium oxysporum* f. sp. *cubense*	*Foc*-milR138	*MaLYK3*	Pathogen-host	[94]

## 4. Applications of miRNA in Crop Protection and Yield Enhancement

In the complex process of plant–pathogen interactions, miRNAs play a crucial role in crop protection and yield increase. With the rapid development of biotechnology, a series of miRNA-based biotechnological tools, such as miRNA overexpression, miRNA silencing technology, and CRISPR/Cas9-mediated precise miRNA editing technology, have been developed and applied, bringing new opportunities for enhancing crop resistance to pests and diseases and increasing yields. These technological breakthroughs not only deepen our understanding of the functional mechanisms of miRNAs but also show great potential in the fields of crop breeding and green pest control. From the construction of traditional overexpression vectors to precise gene editing, miRNA-based biotechnology is gradually establishing a multi-level and multi-dimensional crop protection and yield increase system, providing a new technological path to address the global food security challenge.

### 4.1. Biotechnological Applications Based on miRNA Overexpression

The construction technique of miRNA overexpression vectors, with its controllability and high efficiency, has become a core strategy for enhancing resistance and yield in the field of crop genetic improvement. This technique involves introducing exogenous miRNA expression elements into target crops. A promoter is used to drive the continuous transcription of miRNA precursors (pre-miRNA). Through the plant’s endogenous processing mechanism, mature miRNAs are formed, which can then target and regulate the expression of downstream genes based on the principle of base complementary pairing. This technique has a certain universality among different crop species. miR171b plays a positive regulatory role in citrus resistance to Huanglongbing [51]. Bacteria could be detected in the control plants as early as the second month after infection. While in the overexpressing plants, bacteria were not detected until the 24th month. The plants overexpressing miR171b were more disease-resistant. These studies not only verified the feasibility of miRNA overexpression technology in crop improvement but also provided a theoretical basis for pathogen control and directed breeding by analyzing the molecular mechanisms. With the development of synthetic biology technology, modular vector design (such as the Golden Gate assembly system [99]) can rapidly construct polycistronic miRNA expression units to regulate multiple target pathways simultaneously, opening a new path for cultivating crop varieties with both high yield and multiple resistances.

### 4.2. Applications of miRNA Silencing Technology

The short tandem target mimic (STTM) technology is an important and efficient means to silence the activity of endogenous miRNAs. Its core principle is based on the base complementary pairing mechanism between miRNAs and target mRNAs. By constructing an expression vector containing tandemly repeated miRNA target mimic sequences, when these mimic sequences pair complementarily with miRNAs, due to the specific mismatch design, they will not be degraded by the miRNA-mediated cleavage, but can competitively bind to miRNAs, thus blocking the binding of miRNAs to natural target mRNAs and effectively inhibiting the activity of miRNAs. The activity of endogenous miRNAs can be efficiently silenced by constructing a vector containing tandemly repeated miRNA target mimic sequences [100]. When the STTM technology was used in rice to reduce the expression of 35 important miRNA families, new functions were discovered, such as miR172 affecting stem development and panicle density and miR156 affecting root development. Meanwhile, regulating the STTM expression of miR398 can increase the number of grains per panicle and the panicle length in rice, providing new targets and technological approaches for the genetic improvement of rice [101].

### 4.3. CRISPR/Cas9-Mediated Precise miRNA Editing Technology

The CRISPR/Cas9 system is an important tool in miRNA research and crop improvement due to its ability to edit genes with high precision and efficiency. By designing a guide RNA (gRNA) complementary to a specific region of the plant miRNA gene, the gRNA can precisely target the miRNA gene’s target site after binding to the Cas9 protein. The Cas9 protein exerts its nuclease activity, which causes a double-stranded break (DSB) in the DNA. Non-homologous end joining (NHEJ) or homologous recombination repair (HDR) mechanisms are then activated within the cell. The NHEJ repair process is prone to introducing insertion or deletion mutations (indels), resulting in loss of gene function. HDR, on the other hand, requires the provision of a homologous repair template and can achieve precise gene editing, such as specific base substitutions and fragment insertions, enabling precise regulation of the miRNA gene. The CRISPR/Cas9 system was used to simultaneously knock out miR482b and miR482c in tomato, resulting in two transgenic plants with simultaneous silencing of miR482b and miR482c, and one transgenic plant with silencing of miR482b alone [102]. It was found that the symptoms of late blight were alleviated in the transgenic plants, and their disease resistance was improved. In addition, the simultaneous knockout was more effective than the sole knockout of miR482b, demonstrating the efficacy of using CRISPR/Cas9 to edit miRNAs for the breeding of disease-resistant tomatoes.

## 5. Perspectives

Given the escalating global population, decreasing availability of arable land, and increasing demands for food security worldwide, the development of innovative, sustainable, and eco-friendly strategies for plant disease management has become essential. Double-stranded RNAi technology offers potential advantages for disease control. RNAi is a conserved mechanism for regulating gene expression in organisms. When double-stranded RNA (dsRNA) enters an organism, it is cleaved by nuclease into siRNA. The siRNA binds to the Argonaute (AGO) protein to form a functional RNA-induced silencing complex (RISC) in the host. Through sequence-specific complementarity, the RISC-loaded siRNA guides the recognition and cleavage of target mRNAs [80]. Both endogenous miRNAs and exogenously induced RNAi rely on sequence-specific interactions between small RNAs and their targeted genes. The highly conserved mechanism serves as the basis for the artificial design of RNA molecules for the trans-regulation of gene expression. For prokaryotes, such as bacteria, bacteria lack core RNAi components such as Dicer and RISC and do not have such a complete classical RNAi pathway. However, there are mechanisms with similar functions to RNAi in bacteria. The CRISPR-Cas system in bacteria can recognize and cleave foreign nucleic acids (e.g., phage DNA and plasmid DNA). It complement-pairs with the target nucleic acid via CRISPR RNA (crRNA) and directs the Cas protein to carry out the cleavage, which can be regarded as a kind of “gene silencing” defense mechanism for bacteria to resist the invasion of foreign nucleic acids [103]. For example, in *Streptococcus pyogenes*, the Cas9 protein of the type II CRISPR-Cas system precisely cleaves exogenous DNA complementary to crRNA with the help of crRNA and trans-activating crRNA (tracrRNA) [104]. In recent years, studies have shown that bacteria can artificially regulate gene expression in a manner similar to RNAi [105]. For example, with the help of carriers such as exosomes, siRNAs can be delivered into bacterial cells such as *Escherichia coli* and *Staphylococcus aureus*. These exogenous siRNAs can downregulate the expression of bacterial target genes. For example, in *E. coli*, siAda delivered by exosomes causes a dose-dependent down-regulation of Ada protein expression mainly through translation inhibition [106]. Overall, bacteria do not possess the classic RNAi phenomenon found in eukaryotes. However, they have other mechanisms that can perform functions similar to the regulation of gene expression or resistance to foreign nucleic acids, and gene silencing with effects similar to RNAi can be achieved by artificial intervention.

Two highly promising methods of plant disease control, derived from RNAi technology and the trans-kingdom transfer of miRNA regulatory mechanisms, are HIGS and spray-induced gene silencing (SIGS).

HIGS technology utilizes *Agrobacterium*-mediated transformation to stably express pathogen-targeting dsRNAs in plant cells. During pathogen infection, these dsRNAs are translocated into microbial cells, where they initiate the RNAi pathway. Subsequently, Dicer-dependent processing generates siRNAs, which load onto AGO-containing effector complexes. These complexes mediate the sequence-specific degradation of virulence-related mRNAs, leading to transcriptional suppression of pathogenic determinants [87,107]. Through HIGS technology, the pathogenic genes or effectors of the powdery mildew fungus (*Blumeria graminis*) can be targeted using RNAi to significantly inhibit the formation of fungal haustoria and mycelium development. This study confirms, for the first time, that HIGS can target and inhibit the expression of pathogenic genes of obligate biotrophic fungi [108,109]. Moreover, this method has also achieved remarkable results in the prevention and control of the necrotrophic *Magnaporthe oryza*e (rice blast fungus). HIGS treatment was carried out on six genes, namely *CRZ1*, *PMC1*, *MAGB*, *LHS1*, *CYP51A,* and *CYP51B*, which play important roles in the pathogenicity and development of the pathogen [110]. The HIGS vector was transferred into rice by Agrobacterium-mediated transformation (ATMT), and transgenic rice plants were successfully produced. Except for the *PMC1* and *LHS1* genes, silencing of the other genes significantly reduced the pathogenicity of *Magnaporthe oryzae*. Plant-expressed double-stranded RNA (dsRNA) can mimic the inhibitory effect of plant miRNAs on fungal target genes or interfere with the synthesis and function of fungal milRNAs. Compared to traditional gene editing, HIGS does not require direct modification of the fungal genome. Stable expression of dsRNAs can be achieved through ATMT, viral vectors, or transgenic plants. It is particularly suitable for fungal pathogens that are difficult to transform genetically. By synthesizing fluorescence-labeled RNA in vitro, *Sporisorium scitamineum* was able to take up dsRNA and siRNA and significantly reduce the expression levels of target genes. The *SsGlcP* gene, which plays an important role in the infection process of *S. scitamineum*, was selected as a target, and an RNAi strain was constructed. The infection rate of this strain on sugarcane was slowed, and the disease incidence was reduced. A transgenic sugarcane with the *SsGlcP* gene as a target for HIGS was successfully constructed. After inoculation with *S. scitamineum*, it was found that the *SsGlcP* gene was successfully silenced in the transgenic lines, and the incidence of smut disease in the transgenic sugarcane was significantly reduced [111].

The regulatory effects of milRNAs on plant targets can be tested in reverse by using HIGS to interfere with key genes involved in the biosynthesis of fungal milRNAs, such as *DCL* and *AGO*. For example, silencing the *FgDCL1* of *F. graminearum* can block the production of its milRNAs, thereby revealing the function of milRNAs in suppressing plant immunity [112].

In addition, “double silencing” systems can be constructed to develop novel disease-resistance strategies by designing dsRNAs that target milRNA precursors or key regulators and combining them with the synergistic effects of plant miRNAs. For example, simultaneous interference with the pathogenesis-related sRNAs of *B. cinerea* and the hijacked miRNAs in plants can break the cross-kingdom regulatory advantage of the pathogen.

SIGS technology synthesizes in vitro dsRNAs targeting key genes of pests and diseases, encapsulates them in nanovesicles, and then sprays them onto the plant surface. These dsRNAs can be absorbed by plant tissues through leaves or wounds. They are subsequently taken up by pest cells during feeding or infection, triggering an RNAi effect that silences the target genes [113,114]. In contrast to HIGS, SIGS avoids the transgenic process and offers the advantages of flexible application and relatively low environmental risks. SIGS has been explored for the prevention and control of various plant pests and diseases, including wheat powdery mildew, tomato late blight, and others [115,116,117]. The application of specific dsRNAs effectively inhibits pathogen growth and reduces disease severity.

Artificial miRNA (amiRNA) targets the key genes of pathogens by mimicking endogenous plant miRNAs. It induces the degradation of their mRNAs or translational inhibition, thereby blocking the proliferation or pathogenic process of pathogens. Off-target effects on host genes can be reduced through the precise design of target sequences in the conserved regions of pathogens. In addition, amiRNA meets the needs of sustainable agriculture, as it does not leave harmful residues and is degradable compared to chemical pesticides. Currently, developing miRNA-based biopesticides has become an important approach to controlling future pests and diseases [118].

The target genes of miR-9b and miR-VgR are ABCG4 and VgR, and these target genes are involved in aphid growth and development [119]. Spraying chemically synthesized single-stranded RNA (syn-ssRNA) significantly reduced the survival of peach aphids and increased the rate of malformation. In spray-induced gene silencing (SIGS), the selection of the appropriate miRNA according to the desired target trait in the target organism is the key to the successful application of this technology for pest and disease control. Based on this, websites such as WMD3-Web MicroRNA Designer [120] and amiRNA Design Helper [121] can be used to design miRNAs that specifically target the genes associated with pathogenicity or toxin production of pathogens. Spraying amiRNA on the surface of plants can inhibit the infection of pathogens and prevent their colonization in plants, thus protecting plants from pathogen attacks. In the process of cross-kingdom regulation of gene expression, both miRNAs and milRNAs can be delivered via EVs. To improve the delivery efficiency, researchers developed nanoparticle (NP) carrier systems based on mesoporous silica nanoparticles (MSN) [122] or poly(lactic-co-glycolic acid) (PLGA) [123]. By co-encapsulating plant-derived EVs and amiRNA mimics within NPs, the stability, delivery efficiency, and cross-species regulatory capacity of the SIGS system can be significantly improved, providing an innovative solution for fungal disease control.

In addition to HIGS, SIGS technologies, and amiRNA, the development of miRNA transgenic plants represents another highly promising strategy for improving plant disease resistance. Key miRNA-encoding genes are introduced into the plant genome through genetic engineering to enable stable miRNA expression. Transgenic rice plants overexpressing miR160a and miR398b exhibited increased resistance to *M. oryzae* [124]. Compared to traditional disease control methods, the development of miRNA transgenic plants is highly specific and has minimal impact on the normal physiological functions of plants. Moreover, it reduces fungicide use and environmental pollution, aligning with the concept of sustainable agricultural development. To date, several studies have been conducted on rice, vines, and other plants [125,126]. Although significant progress has been made in the trans-kingdom regulation of miRNAs, numerous unknown areas remain to be explored. On the one hand, it is essential to further investigate miRNAs involved in various plant–pathogen interactions and to elucidate their targets and regulatory networks. On the other hand, it is crucial to thoroughly analyze the molecular mechanisms underlying miRNA translocation between plants and pathogens, including the formation of transporter carriers, miRNA sorting, and the stability and functionality of miRNAs post-translocation. Furthermore, a key direction for future research will be translating basic research results into practical applications and developing green, efficient, and sustainable biocontrol strategies based on miRNAs. For instance, the construction of transgenic plants stably expressing specific miRNAs, along with the design of synthetic miRNA analogs, will offer novel ideas and methods for preventing and controlling fungal diseases in agricultural production.

## Figures and Tables

**Figure 1 plants-14-01250-f001:**
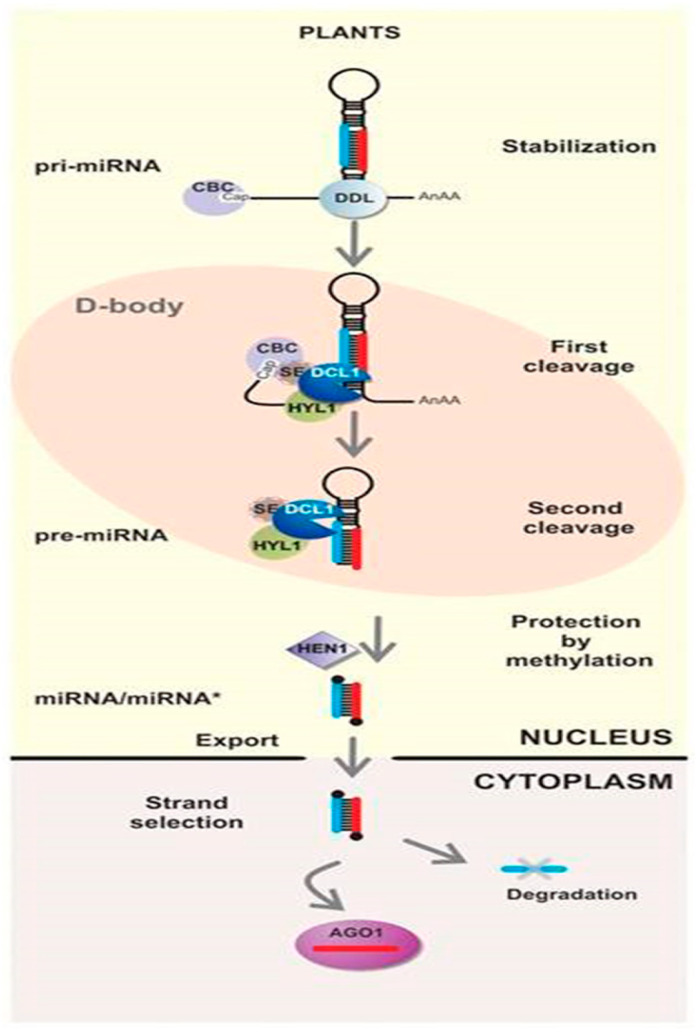
Schematic diagram of miRNA biogenesis and function in plants [27]. The biogenesis pathway of miRNA in plants. The biogenesis of miRNAs in plants typically occurs within the Dicing body (D-body), which consists of a cleavage complex made up of the Cap Binding Complex (CBC), the zinc finger protein SERRATE (SE), the double-stranded RNA binding protein HYPONASTIC LEAVES 1 (HYL1), and Dicer-like 1 (DCL1). Among them, DCL1 performs two cleavage events on the pri-miRNA to generate an miRNA/miRNA* duplex. Subsequently, HUAENHANCER1 (HEN1) recognizes the miRNA/miRNA* duplex and methylates the last nucleotide at the 3′ end of each strand of the duplex to maintain its structural stability. Finally, HASTY (HST) transports the duplex out of the nucleus. In the cytoplasm, the miRNA/miRNA* duplex is recognized by Argonaute (AGO) proteins to form the miRNA-induced silencing complex (miRISC). The RNA helicase within miRISC separates the miRNA/miRNA* duplex. Thereafter, the miRNA regulates the expression of target genes while the miRNA* is rapidly degraded.

**Figure 2 plants-14-01250-f002:**
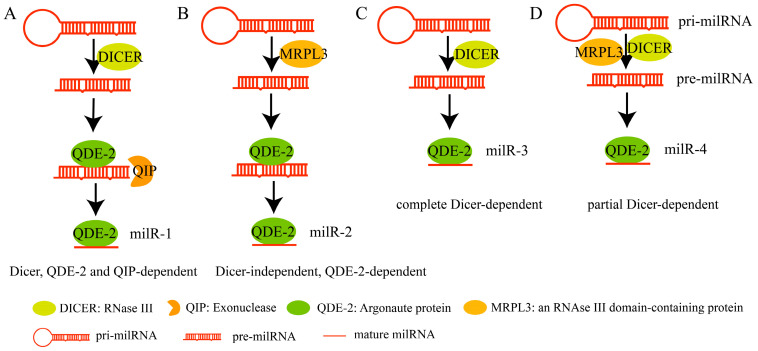
Biogenesis of fungal milRNAs: (**A**) Biogenesis of milR-1 is dependent on Dicer, QDE-2, and QIP. (**B**) The biogenesis mechanism of milR-2 is independent of Dicer and dependent on MRPL3 and QDE-2. (**C**) The synthesis of milR3 relies entirely on DCL proteins, similar to miRNA synthesis in plants. (**D**) The biogenesis of milR-4 is partially dependent on Dicer.

**Figure 3 plants-14-01250-f003:**
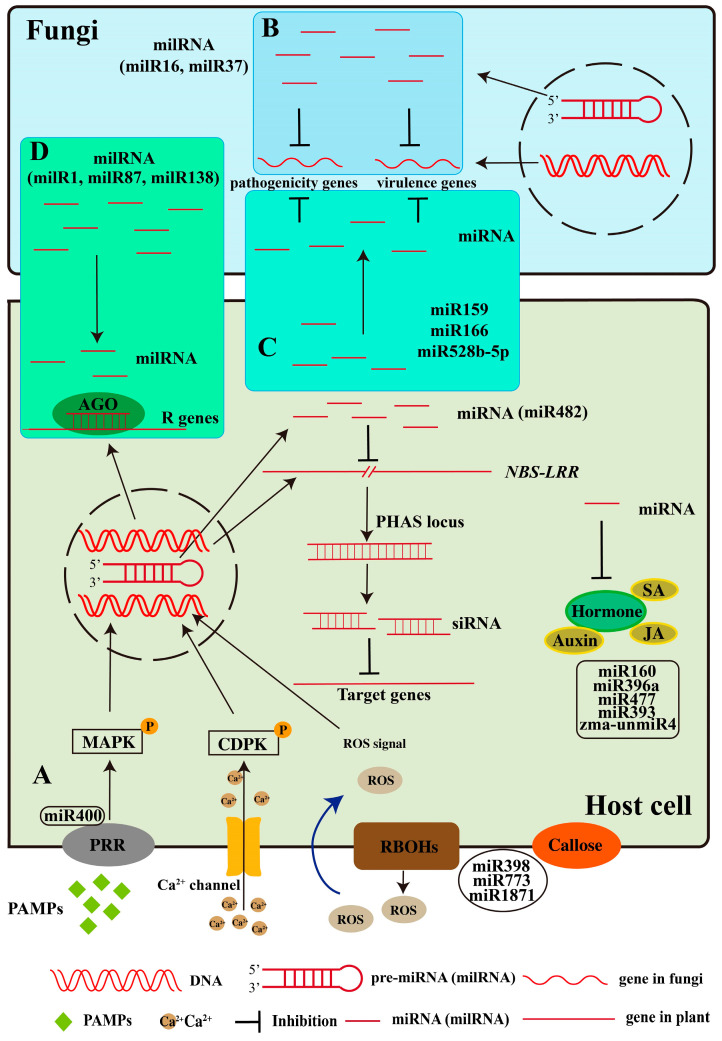
miRNAs (milRNA) are involved in plant–pathogen interaction: (**A**) Plant-derived miRNAs regulate endogenous gene expression by binding to target mRNAs through base-complementary pairing. These miRNAs are involved in regulating various biological processes in response to PTI and ETI, including ROS accumulation, callose deposition in the cell wall, rapid Ca^2+^ influx through ion channels, fine regulation of phytohormone signaling pathways, and expression of defense-related genes. Callose deposition in the cell wall, rapid Ca^2+^ endocytosis through ion channels, fine regulation of phytohormone signaling pathways, and the expression of defense-related genes ultimately play a crucial role in plant disease resistance. (**B**) milRNAs are involved in pathogenicity by specifically targeting and regulating the expression levels of endogenous genes related to pathogenicity and virulence. (**C**) Trans-kingdom regulation occurs in plant–pathogen interactions. On the one hand, plant miRNAs can be transferred to fungi to reduce pathogen pathogenicity by targeting and inhibiting the expression of pathogenicity- or virulence-related genes in fungi, thus decreasing pathogen virulence. (**D**) On the other hand, fungal miRNAs can be transferred to plants and reduce plant disease resistance by interfering with the expression of disease resistance genes.

**Figure 4 plants-14-01250-f004:**
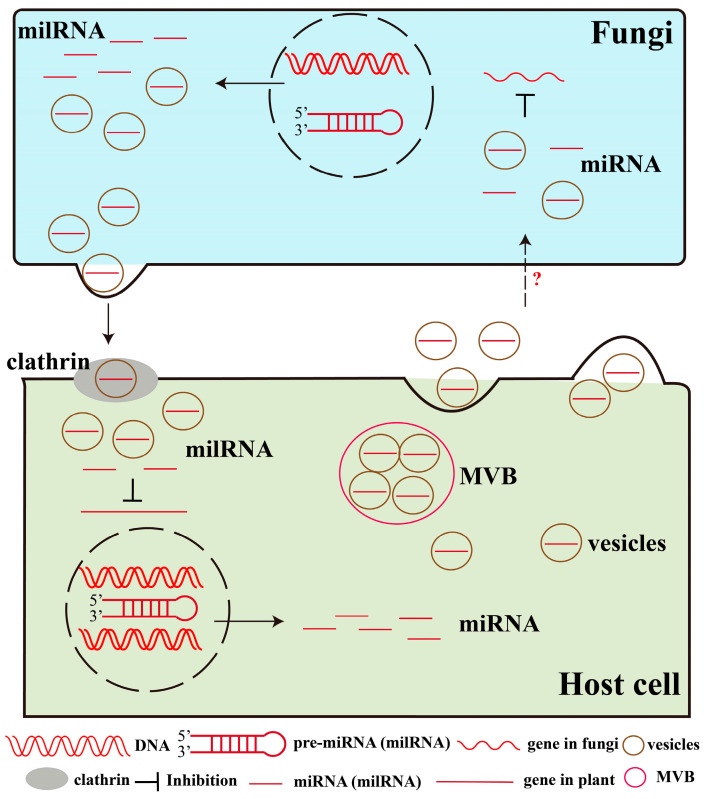
The secretion pathway of miRNAs (milRNAs) during the interaction between host plants and pathogens. Plant miRNAs can be translocated to fungi via extracellular vesicles (EVs) during pathogen infection. They are released after the fusion of multivesicular bodies (MVBs) with the cell membrane. However, it is not yet clear how the miRNAs in the EVs get into the fungal cells. Fungi milRNAs are transported outside of plant cells by EVs. Clathrin-mediated endocytosis (CME) in plants is able to take up EVs containing milRNAs in order to enter the plant cell and regulate the expression of their target genes.

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
