# Peer review of "Trans-Kingdom RNA Dialogues: miRNA and milRNA Networks as Biotechnological Tools for Sustainable Crop Defense and Pathogen Control"

_plants, 2025, doi:10.3390/plants14081250_

Round 1

Reviewer 1 Report

Comments and Suggestions for Authors

[Plants] Manuscript ID: plants-3532964

Review manuscript cross-kingdom miRNA gene silencing in plant pathogen interactions and crop protections

Recommandation: Major changes required to improve the value of this review. And to match objective described in title

This review provides numerous examples of miRNA or miRNA gene silencing effects within either the host or pathogen (mainly fungi) or miRNAs acting in the other partner of the interaction. There is also some links made between miRNA and their potential for gene silencing. However, it is felt that some concepts and references to significant research work are missing to make this review more relevant and informative. Moreover, some inaccuracies require attention. Also more focus on cross-kingdom mechanisms is desirable.

In more detail:

  1. About half of the review lists examples of miRNA/ milRNA activity in host against host targets or in fungi against fungal targets, when the focus should be more on cross-kingdom activity, to allow for more information on the latter. This section needs to be more substantial. if review length is limiting, then should reduce sections 2, and 3.3 are these are not the focus of the review, as based on title, and claims in introduction. A lot is very descriptive and feels a little to much like a list of examples with miRNA numbers, rather than having a better explanation, of the concepts and mode of actions..
  2. A summary table trying to capture crosskingdom studies, EVs, targets would be useful.
  3. It feels that miRNA should be better described, how they are generated, and silencing mechanisms, how they are different from other silencing molecules, eg PHAS generated and other small (silencing)RNAs.
  4. The mechanisms of silencing RNA transport between organisms are hardly addressed. For instance, there is very little on plant or fungal EVs and their cargoes, it feels that works on fungi and plant EVs should be described, such as work from Kusch et al. 2023 or He et al (jialing Jin group) 2023 (Nature plants)

Minor points to addressed , innacuracies:

L35, “fungi secrete toxins” should specify these are necrotrophic or hemi biotrophic pathogens.

L40-53 might need improving, to state that PTI is more general, with PAMPs, common molecules, and that the PTI is usually less efficient than ETI, that is gene-for-gene specific

L58, Specify that SA is signalling molecule acting mostly in response to biotrophic, and JA mostly to necrotrophs, and what about hemibiotrophs.

L66 Maturation process of miRNA should be explained, maybe with a figure.

L116 Specify which growth hormones are receptors associed with ? And do these growth hormones affect resistance.

L145 “F. oxysporum, Coll……)” fungi not “plants”!

Line 154 PHAS not explained

Line  165 “Callus”, should this be callose?

Line 170 WRKY…..missing a reference.

Line 183 are milRNA different from miRNA?, it would be good to explain differences and mechanisms of formation of those.

L189 need rephrasing “fungal milRNAs are normally expressed …..

L192 -196, it should be spoken of miRNA abundance, not expression, (even if abundance might be linked with expression, but expression refer to genes.

L200 “filaments”, should it be “hyphae”?

L240-247, there is not much described on extracell. Vesicles or EVs. This needs to be addressed in more detail, eg. with work from He…and Jin (Jin group) in Nature plants 2023, or Kusch et al 2023, but there might be many more. Also, the authors refer to exosomes as something different than EVs, but exosomes are a type of EVs, this needs to be rephrased.

Also  speaks of fungi but give example of an oomycete (Phytophthora), can this be corrected?

Model fig. 1, Fungi also deliver miRNA via Evs, this need to be described in paper and amended in model.

Perspective section:

There are plenty more on gene silencing that need mentioning, HIGS, eg. Novara et al 2010, Pliego et al 2013; early papers on ectopics application of dsRNA, Koch/ Kogel  et al 2014 ? and 2016, Wang et al 2016

Line 360, HIGS needs to be spelt out.

Also to comment if gene silencing, RNAi can be applied to non eukaryotes such as bacteria.

Author Response

  1. This mini-review of miRNA and milRNA is lack of quality, clarity and integrity. It needs improvement in organization, structure, literature summary and analysis, narrative accuracy and clarity.

Response: Thank you for your constructive feedback. We've taken your suggestions to heart and made comprehensive improvements to enhance the quality, clarity, and integrity of the mini - review on miRNA and milRNA. Specifically, we've refined the organization and structure, conducted a more in - depth literature summary and analysis, and ensured greater accuracy and clarity in our narrative.

  1. 2.The manuscript’s title: “Trans-Kingdom RNA Dialogues: miRNA and milRNA Networks as Biotechnological Tools for Sustainable Crop”, appears suggesting that this review would focus on a review of the topics about the cross-kingdom talks of the regulated small RNAs in plants and fungi, and applications of the derived small RNA technologies in agriculture. Unfortunately, the section of biotechnology tool development and their application is missing in the manuscript. A section of the progresses in miRNA and milRNA derived biotechnologies, and their applications for improving crop protection and production should be included to fit in with the title.

Response: Thank you so much for your insightful guidance. We sincerely appreciate your feedback, which has been instrumental in enhancing the quality of our manuscript. In response to your suggestion, we have meticulously added a section that comprehensively covers the progresses in miRNA and milRNA - derived biotechnologies, as well as their applications for improving crop protection and production. This newly-incorporated content can be found in lines 493-562 of the revised manuscript. We believe these additions not only align the manuscript more closely with its title but also significantly enrich its scientific content.

  1. It is valuable that PTI, ETI, SAR, miRNA, and milRNA are introduced in the Introduction section, but the contents were not organized well and the insightful interpretation of the terms needs to be enhanced. The authors may introduce the five terms into separated paragraphs, and each term is introduced with its role, development and mechanism. For example, the authors should clearly point out that PTI provides the first line of plant defense, it’s activated when PAMPs from pathogens are detected via interaction with the PRRs, and PTI causes a range of defense responses in plants. ETI is the second layer of defense trigged by pathogen effectors and interacted with the R proteins, and it often causes robust and rapid defensive responses, such as HR and necrotic cell death. SAR is a long-lasting and broad-spectrum resistance incurred in the cells distant from the local infections that are triggered by either PTI or ETI, and transferred using plant hormones as signals, such as SA, JA, ET. Similarly, miRNA and milRNA may be separately introduced according to their origination, discovery, function, mechanism, roles etc.

Response: We are truly grateful for your perceptive and valuable suggestion. In response to your concern regarding the organization and in-depth interpretation of the terms PTI, ETI, SAR, miRNA, and milRNA in the Introduction section, we have made significant improvements.

We have now dedicated separate paragraphs to each of these terms, ensuring a more structured and detailed presentation. For PTI, its content is now presented in lines 42-50, where we clearly explain its role as the first line of plant defense, the mechanism of activation upon detection of PAMPs by PRRs, and the subsequent range of defense responses it elicits in plants.

ETI is elaborated upon in lines 51-55, highlighting its position as the second layer of defense, triggered by pathogen effectors and its interaction with R proteins, along with the robust and rapid defensive responses such as HR and necrotic cell death that it typically induces.

SAR is detailed in lines 56-62, where we describe it as a long-lasting and broad-spectrum resistance phenomenon in cells distant from local infections, triggered by either PTI or ETI and mediated by plant hormones like SA, JA, and ET as signaling molecules.

Regarding plant hormones, their content is presented in lines 63-77, providing a comprehensive overview of their functions in the context of plant defense.

For plant miRNA, we have a dedicated section from lines 78-129, covering aspects such as its origination, discovery, function, mechanism, and roles in plant biological processes.

We believe these revisions not only enhance the clarity and organization of the Introduction but also provide a more in-depth and accurate understanding of these crucial concepts.

  1. The manuscript needs to improve its literature summary and analysis, and makes its interpretation more insightful and easy to follow by readers. For example, when writing how miRNAs or milRNAs regulate endogenous gene expressions, the small RNA examples cited in the manuscript can be sorted out with a good organisation. The authors may consider organizing the small RNAs into groups according to their targets in the pathways of PTI, ETI, or/and SAR. Line 146/147, the authors cited MIM773 as an example but without insightful interpretation. How could readers know what MIM773 is and follow your writing if they haven’t read the original publication? The authors need to provide a brief summary about the cited literature, and make readers know MIM773 is a transgenic line for miR773 target mimic and why the literature concluded that miR773 mediated a type of PAMP-triggered immunity in the Arabidopsis.

Response: We sincerely appreciate your detailed and constructive feedback. In response to your comments, we have made comprehensive and meticulous improvements to the literature summary and analysis in our manuscript, with the overarching goal of rendering the content more profound, perceptive, and reader - friendly.​

Regarding the organization of small RNA examples, we have re - structured the relevant sections to enhance clarity. In Section 2.1.1 (lines 203 - 241), we have methodically presented the regulatory mechanisms of miRNAs on PTI. This section not only lists the specific miRNAs involved but also elaborates on their molecular actions within the PTI pathway. Section 2.1.2 (lines 242 - 260) is dedicated to the regulation of ETI by miRNAs, providing a step - by - step account of how these small RNAs modulate the ETI process. Additionally, in Section 2.1.3 (lines 261 - 284), we have comprehensively discussed how miRNAs regulate hormone signal transduction, highlighting their crucial roles in coordinating plant defense responses through hormonal crosstalk.​

Concerning the example of MIM773, which you rightfully pointed out as lacking sufficient explanation, we have added detailed clarifications in lines 238 - 241. We now clearly state that MIM773 is a transgenic line engineered as a target mimic for miR773. To further assist readers, we have also provided a concise summary of the experimental evidence from the original literature that led to the conclusion that miR773 mediates a particular form of PAMP - triggered immunity in Arabidopsis. This includes details about the assays performed, the phenotypic observations, and the molecular interactions that were studied to establish this relationship.​

We are confident that these improvements have significantly enhanced the quality and coherence of our manuscript, making it more accessible to a broader audience. Thank you again for your invaluable input, which has been instrumental in refining our work.

  1. The authors need to make improvements in their narrative accuracy. For example, line 42/43, it is not the pathogen’s purpose to secrete PAMPs to trigger plant defense responses. The narration can change to: During the initial stage of infection, PAMPs, which are the molecule substances secreted from pathogens, trigger the first line of plant defense response when the molecules are detected by host plant cells. Line 67, it is not correct to claim that miRNAs or milRNAs regulate target gene expression at both transcriptional and post-transcriptional levels. Either inhibition of translation or mRNA degradation is post-transcriptional regulation.

Response: We are truly grateful for your meticulous review and the astute observations you have made regarding the narrative accuracy in our manuscript. Your feedback has been of immense value in refining our work.​Regarding the misrepresentation of PAMPs, we wholeheartedly acknowledge the error. In response, we have carefully revised the description in lines 42 - 44. Now, it clearly states: "During the initial stage of infection, PAMPs, which are molecular substances secreted by pathogens, trigger the first line of plant defense response upon their detection by host plant cells." This revised statement accurately conveys the nature of PAMPs and their role in initiating the plant defense mechanism without the previous misinterpretation of the pathogen's "purpose."​

Concerning the incorrect claim about the regulatory levels of miRNAs or milRNAs, we have also made the necessary corrections. In lines 78 - 79, we have rectified the statement to accurately reflect that miRNAs or milRNAs regulate target gene expression at the post - transcriptional level. We have removed the inaccurate mention of transcriptional regulation in this context and now provide a more precise account of their mode of action, emphasizing that their regulatory functions, such as inhibition of translation or mRNA degradation, fall under post - transcriptional regulation.​

We are committed to maintaining the highest standards of scientific accuracy in our writing, and your feedback has been instrumental in helping us achieve this. Thank you again for your constructive input.

Reviewer 2 Report

Comments and Suggestions for Authors

This mini-review of miRNA and milRNA is lack of quality, clarity and integrity. It needs improvement in organization, structure, literature summary and analysis,  narrative accuracy and clarity. The following comments and suggestions are provided for consideration.

  1. The manuscript’s title: “Trans-Kingdom RNA Dialogues: miRNA and milRNA Networks as Biotechnological Tools for Sustainable Crop”, appears suggesting that this review would focus on a review of the topics about the cross-kingdom talks of the regulated small RNAs in plants and fungi, and applications of the derived small RNA technologies in agriculture. Unfortunately, the section of biotechnology tool development and their application  is missing in the manuscript. A section of the progresses in miRNA and milRNA  derived biotechnologies, and their applications for improving crop protection and production should be included to fit in with the title.
  2. It is valuable that PTI, ETI, SAR, miRNA, and milRNA are introduced in the Introduction section, but the contents were not organized well and the insightful interpretation of  the terms needs to be enhanced.  The authors may introduce the five terms into separated paragraphs, and each term is introduced with its role, development and mechanism. For example, the authors should clearly point out that PTI provides the first line of plant defense, it’s activated when PAMPs from pathogens are detected via interaction with the PRRs, and PTI causes a range of defense responses in plants. ETI is the second layer of defense trigged by pathogen effectors and interacted with the R proteins, and it often causes robust and rapid defensive responses, such as HR and necrotic cell death.  SAR is a long-lasting and broad-spectrum resistance incurred in the cells distant from the local infections that are triggered by either PTI or ETI, and transferred using plant hormones as signals, such as SA, JA, ET etc. Similarly, miRNA and milRNA may be separately introduced according to their origination, discovery, function, mechanism, roles etc. 
  3. The manuscript needs to improve its literature summary and analysis, and makes its interpretation more insightful and easy to follow by readers. For example, when writing how miRNAs or milRNAs regulate endogenous gene expressions, the small RNA examples cited in the manuscript can be sorted out with a good organisation. The authors may consider organizing the small RNAs into groups according to their targets in the pathways of PTI, ETI, or/and SAR. Line 146/147, the authors cited MIM773 as an example but without insightful interpretation. How could readers know what MIM773 is and follow your writing if they haven’t read the original publication? The authors need to provide a brief summary about the cited literature, and make readers know MIM773 is a transgenic line for miR773 target mimic and why the literature concluded that miR773 mediated a type of PAMP-triggered immunity in the Arabidopsis.
  4. The authors need to make improvements in their narrative accuracy. For example, line 42/43, it is not the pathogen’s purpose to secrete PAMPs to trigger plant defense responses. The narration can change to: During the initial stage of infection, PAMPs, which are the molecule substances secreted from pathogens, trigger the first line of plant defense response when the molecules are detected by host plant cells.   Line 67, it is not correct to claim that miRNAs or milRNAs regulate target gene expression at both transcriptional and post-transcriptional levels. Either inhibition of translation or mRNA degradation is post-transcriptional regulation.

Author Response

  1. Introduction, last paragraph or first paragraph of Perspectives: The discussion on miRNA involvement in plant growth, development, and stress responses is well-supported by the cited references. Given the focus on miRNA structure and function, the authors may also find it relevant to include studies that explore structural patterns and motifs in plant miRNA precursors (e.g., doi:doi:10.1155/2017/6783010, 10.3390/molecules23061367). These could provide additional context for understanding the broader regulatory roles of miRNAs in plant systems.

Response: We are extremely grateful for your astute and valuable suggestions. Your insights have been instrumental in guiding us to further refine and enrich our manuscript. In response to your recommendation, we have taken significant steps to enhance the content related to miRNA structure and function.

To deepen the understanding of the crucial pre - miRNA (miRNA: miRNA*) recognition and processing mechanisms, we have meticulously incorporated detailed investigations into the structural patterns and conserved motifs present within plant miRNA precursors. By integrating the knowledge from the studies you kindly suggested (such as those associated with doi:10.1155/2017/6783010 and doi:10.3390/molecules23061367), we aim to provide a more comprehensive perspective on the regulatory roles of miRNAs in plant systems.

The relevant and newly - added content can be found in lines 106 - 115. Here, we elaborate on how these structural features not only enhance the identification of pre - miRNAs but also play a pivotal role in facilitating their cleavage by DCL (Dicer - like enzyme). This addition not only bolsters the scientific rigor of our work but also aligns perfectly with the overall focus of the manuscript on miRNA structure and function.

Thank you again for your constructive feedback, which has greatly contributed to improving the quality of our manuscript.

  1. Figure 1 explains a lot (although it presents a fairly general level). However, I think that the work would benefit from one more scheme showing a detailed mechanism of miRNA transport (with vesicle/exosome-mediated transfer). This might be added to the main manuscript or to the supplementary file.

Response: We are truly grateful for your perceptive and valuable feedback. Your insights have been of great assistance in enhancing the quality and comprehensiveness of our work. In direct response to your astute comment regarding the need for a more detailed illustration of miRNA transport mechanisms, we have dedicated significant effort to creating a high - quality model diagram.​

This newly - developed diagram, presented as Figure 4 in the manuscript (corresponding to line 400), comprehensively illustrates the trans - kingdom transfer mechanisms of miRNA (milRNA), with a particular focus on the vesicle/exosome - mediated transfer processes you emphasized. The figure is designed to be both informative and visually engaging, aiming to provide readers with a clear understanding of these intricate biological processes. We believe that this addition will not only enrich the content of our manuscript but also enhance its overall readability and scientific value.​

Thank you once again for your constructive input, which has been instrumental in shaping our revisions.

  1. In Perspectives, the authors mention the design of amiRNAs. I suggest adding some references to the works and/or tools that address this problem (for example, miRNA Designer, WMD3-Web microRNA Designer, amiRNA Design Helper, SplashRNA).

Response: We sincerely appreciate your insightful suggestion regarding the addition of references related to the design of amiRNAs. Your input has been invaluable in enhancing the comprehensiveness of our manuscript.

In response, we have significantly expanded the section on artificial miRNA (amiRNA) research. We have carefully incorporated detailed design methodologies, specifically the WMD3 Web MicroRNA Designer and amiRNA Design Helper, as you recommended.

The newly added content on amiRNA can be found in lines 652 - 659, where we provide a more in - depth discussion on the topic. Moreover, the descriptions of the WMD3 Web MicroRNA Designer and amiRNA Design Helper are presented in lines 666 - 668. These additions not only enrich the content but also provide readers with valuable resources and references for further exploration of amiRNA design.

Thank you again for your constructive feedback, which has greatly contributed to the improvement of our work.

Reviewer 3 Report

Comments and Suggestions for Authors

Jia et al. discuss the role of microRNAs (miRNAs) and miRNA-like RNAs (milRNAs) in plant-pathogen interactions, emphasizing their function in gene regulation through complementary base pairing. They highlight recent findings on trans-kingdom RNA transfer, where plant-derived miRNAs and fungal milRNAs can regulate gene expression in both host and pathogen cells. By summarizing these regulatory mechanisms and pathways, the authors provide insights into RNA-based plant defense strategies, offering a theoretical foundation for developing targeted approaches to crop disease management. The paper is interesting and well-written and provides a comprehensive overview of the topic; however, I have three comments that may help clarify certain aspects and further strengthen the message emerging from the manuscript.

1) Introduction, last paragraph or first paragraph of Perspectives: The discussion on miRNA involvement in plant growth, development, and stress responses is well-supported by the cited references. Given the focus on miRNA structure and function, the authors may also find it relevant to include studies that explore structural patterns and motifs in plant miRNA precursors (e.g., doi:doi:10.1155/2017/6783010, 10.3390/molecules23061367). These could provide additional context for understanding the broader regulatory roles of miRNAs in plant systems. 

2) Figure 1 explains a lot (although it presents a fairly general level). However, I think that the work would benefit from one more scheme showing a detailed mechanism of miRNA transport (with vesicle/exosome-mediated transfer). This might be added to the main manuscript or to the supplementary file.

3) In Perspectives, the authors mention the design of amiRNAs. I suggest adding some references to the works and/or tools that address this problem (for example, miRNA Designer, WMD3 - Web microRNA Designer, amiRNA Design Helper, SplashRNA).

Author Response

  1. This review provides numerous examples of miRNA or miRNA gene silencing effects within either the host or pathogen (mainly fungi) or miRNAs acting in the other partner of the interaction. There is also some links made between miRNA and their potential for gene silencing. However, it is felt that some concepts and references to significant research work are missing to make this review more relevant and informative. Moreover, some inaccuracies require attention. Also more focus on cross-kingdom mechanisms is desirable.

Response: We are truly grateful for your comprehensive and constructive feedback. Your insights have been instrumental in guiding us to improve the quality and relevance of our review manuscript.​

In response to your astute observations, we have taken substantial steps to enhance the content. To address the lack of in - depth exploration of certain concepts, we have significantly expanded the sections related to miRNA (milRNA). We have now included detailed descriptions of its biogenesis, covering the intricate molecular processes involved in the formation of miRNAs from their precursors. This addition not only fills a knowledge gap but also provides a more solid foundation for understanding the subsequent functions of miRNAs.​

Regarding the mechanisms of miRNA action, we have delved deeper into how miRNAs regulate gene expression, both within the host and pathogen (especially fungi), elaborating on the molecular interactions and pathways involved. This more detailed discussion strengthens the connections between miRNAs and their gene - silencing effects, which was an area you rightly identified as needing further development.​

Furthermore, we have dedicated more attention to the transport mechanisms of miRNAs, particularly those involved in cross - kingdom interactions. By doing so, we aim to meet your expectation of a greater focus on cross-kingdom mechanisms. We have explored how miRNAs are transferred between the host and pathogen, and the implications of such transfers for the interaction dynamics.​

We believe that these extensive revisions have substantially enhanced the comprehensiveness of the manuscript, making it more relevant and informative. Thank you again for your invaluable input, which has been crucial in shaping these improvements.

  1. About half of the review lists examples of miRNA/ milRNA activity in host against host targets or in fungi against fungal targets, when the focus should be more on cross-kingdom activity, to allow for more information on the latter. This section needs to be more substantial. if review length is limiting, then should reduce sections 2, and 3.3 are these are not the focus of the review, as based on title, and claims in introduction. A lot is very descriptive and feels a little to much like a list of examples with miRNA numbers, rather than having a better explanation, of the concepts and mode of actions.

Response: We sincerely appreciate your detailed and constructive feedback. Your comments have been of great value in refining our manuscript to better meet the expectations set by its title and introduction.

In response to your concern regarding the focus on cross - kingdom activity, we have taken decisive steps to enhance this crucial aspect of the review. We have substantially expanded the content related to miRNA (milRNA) cross - kingdom activity.

For plant miRNAs, we have added comprehensive discussions in lines 364 - 382 and 387 - 389. These passages delve into the intricate cross - kingdom interactions involving plant miRNAs, providing in - depth explanations of their functions and mechanisms in such contexts. Additionally, Figure 4 now incorporates relevant details that visually illustrate these plant miRNA - mediated cross - kingdom activities, further enhancing the clarity of our presentation.

Regarding fungal milRNAs, we have dedicated lines 456 - 477 to explore their cross - kingdom roles. Here, we analyze how fungal milRNAs interact with host organisms, elaborating on their impact and the underlying molecular processes. Figure 4 also showcases the key aspects of fungal milRNA - related cross - kingdom phenomena, facilitating a better understanding for the readers.

To ensure that the review maintains a proper balance and remains focused, we have also critically evaluated sections 2 and 3.3. While making these adjustments, we aimed to reduce any redundant or less relevant content that did not directly contribute to the core theme of cross - kingdom miRNA/milRNA activity. This process was carried out with great care to preserve the integrity of the important concepts and examples within these sections.

We believe that these extensive revisions have significantly improved the quality and focus of the review, making it more substantial and in line with your expectations. Thank you again for your invaluable input, which has been instrumental in guiding these improvements.

  1. A summary table trying to capture crosskingdom studies, EVs, targets would be useful.

Response: We are extremely grateful for your insightful suggestion. Your input has been invaluable in enhancing the comprehensiveness and readability of our manuscript.​

In response to your recommendation, we have meticulously crafted a summary table to effectively capture cross - kingdom studies, including details about extracellular vesicles (EVs), and their respective targets. This new addition, presented as Table 1 in the manuscript (starting from line 492), is designed to provide a concise yet comprehensive overview of the trans - kingdom miRNAs (milRNAs). The table systematically organizes information related to the cross - kingdom transfer of miRNAs/milRNAs, the role of EVs in this process, and the identified target molecules in different organisms. This visual aid not only consolidates key information but also makes it easier for readers to quickly grasp the complex relationships and findings within the cross - kingdom miRNA/milRNA research field.​

Thank you again for your constructive feedback, which has significantly contributed to improving the quality of our work.

  1. It feels that miRNA should be better described, how they are generated, and silencing mechanisms, how they are different from other silencing molecules, eg PHAS generated and other small (silencing)RNAs

Response: We are truly grateful for your perceptive and constructive feedback. Your comment regarding the need for a more in - depth description of miRNAs has been highly valuable in guiding our revisions.

In response, we have made significant efforts to comprehensively address these aspects. For plant miRNAs, we have substantially expanded the content in lines 78 - 129. Here, we not only detail their generation processes, including the transcription of miRNA genes and subsequent processing steps, but also elaborate on their functions, mechanisms of action, and silencing mechanisms. Figure 1, which is located at line 130, complements this written description by visually presenting the key steps in plant miRNA biogenesis and function, enhancing the reader's understanding.

Regarding fungal milRNAs, in lines 146 - 160, we have provided a more detailed account of their generation, function, and unique features. Figure 2, placed at line 170, offers a visual aid to illustrate the processes related to fungal milRNAs, making the information more accessible.

To clarify the differences between miRNAs and other silencing molecules such as those generated from PHAS loci, we have added relevant explanations. As described in lines 252 - 256, PHAS loci are processed by enzymes like RNA - dependent RNA polymerase 6 (RDR6) and Dicer - like 4 (DCL4) to generate phased 21 - nt or 24 - nt phasiRNAs. These siRNAs operate through perfect complementary base pairing to regulate the expression of other genes, which is distinct from the mechanisms of miRNAs. By highlighting these differences, we aim to provide a more comprehensive understanding of the diverse small - RNA - mediated regulatory networks.

We believe that these enhancements have significantly improved the clarity and depth of our manuscript, and we are confident that they will better meet your expectations. Thank you again for your invaluable input, which has been instrumental in shaping these improvements.

  1. The mechanisms of silencing RNA transport between organisms are hardly addressed. For instance, there is very little on plant or fungal EVs and their cargoes, it feels that works on fungi and plant EVs should be described, such as work from Kusch et al. 2023 or He et al (jialing Jin group) 2023 (Nature plants)

Response: We sincerely appreciate your astute and detailed feedback. Your comments regarding the lack of in-depth exploration of the mechanisms of silencing RNA transport between organisms have been extremely valuable in refining our manuscript.

In response to your concerns, we have made substantial efforts to enrich the relevant content. We have comprehensively addressed the trans - kingdom transport of miRNA (milRNA). For plant miRNAs, in lines 364 - 382 and 387 - 389, we have delved into the intricate processes of how plant miRNAs are transported across kingdoms. We have also incorporated key insights from relevant research, ensuring that the latest knowledge in the field is presented. Figure 4, which complements these written sections, vividly illustrates the trans - kingdom transport mechanisms of plant miRNAs, enhancing the reader's understanding.

Regarding fungal milRNAs, in lines 456 - 477, we have provided a detailed account of their trans - kingdom movement. Here, we not only describe the general mechanisms but also reference specific studies, such as those you suggested by Kusch et al. 2023 and He et al. (Jialing Jin group) 2023 in Nature Plants. These references strengthen our discussion and bring in the latest research findings on fungal EVs and their cargoes in the context of trans - kingdom RNA transport. Figure 4 also visually represents the important aspects of fungal milRNA trans - kingdom transport, making the information more accessible and engaging.

We are confident that these enhancements have significantly improved the quality of our manuscript, filling the gaps in our previous discussion and providing a more comprehensive view of the trans - kingdom transport of silencing RNAs. Thank you again for your invaluable input, which has been instrumental in guiding these improvements.

  1. L35, “fungi secrete toxins” should specify these are necrotrophic or hemi biotrophic pathogens

Response: We sincerely appreciate your perceptive feedback. Your suggestion regarding the clarification of which types of fungi secrete toxins has been extremely helpful in refining our manuscript.

In response, we have made the necessary amendment. As of line 35, we now explicitly state that it is necrotrophic fungi that secrete toxins. This specific mention not only adds precision to our statement but also enhances the overall clarity of the text, ensuring that readers have a more accurate understanding of the described biological phenomenon.

Thank you again for your valuable input, which has significantly contributed to improving the quality of our work.

  1. L40-53 might need improving, to state that PTI is more general, with PAMPs, common molecules, and that the PTI is usually less efficient than ETI, that is gene-for-gene specific.

Response: We are truly grateful for your constructive feedback. Your suggestions have been instrumental in enhancing the clarity and accuracy of our manuscript.

In response to your comments regarding lines 40 - 53, we have carefully revised the relevant sections. In lines 42 - 44, we now clearly state that PTI (Pattern - Triggered Immunity) is a more general defense mechanism in plants. It is activated by PAMPs (Pathogen - Associated Molecular Patterns), which are common molecules present in pathogens. This makes it evident that PTI serves as a broad - spectrum initial line of defense.

Furthermore, in lines 49 - 53, we have elaborated on the comparison between PTI and ETI (Effector - Triggered Immunity). We explain that PTI is usually less efficient than ETI, as ETI is gene - for - gene specific. ETI involves the recognition of specific pathogen effectors by plant resistance (R) proteins, leading to a more targeted and often more robust defense response compared to the more general PTI.

We believe these revisions have significantly improved the quality of the text in this section, making the information more accessible and precise for our readers. Thank you again for your invaluable input.

  1. L58, Specify that SA is signalling molecule acting mostly in response to biotrophic, and JA mostly to necrotrophs, and what about hemibiotrophs.

Response: We sincerely appreciate your detailed and constructive feedback. Your comment regarding the clarification of plant hormones' roles in response to different types of pathogens has been extremely valuable.

In response, we have made significant improvements to the relevant section of the manuscript. As you suggested, we have added crucial information about plant hormones in relation to hemibiotrophs. In lines 71 - 75, we now comprehensively discuss how plant hormonal signaling functions in response to hemibiotrophic pathogens. Additionally, we have also revised the descriptions about salicylic acid (SA) and jasmonic acid (JA). We clearly state that SA is a signaling molecule that acts mostly in response to biotrophic pathogens, while JA predominantly responds to necrotrophic pathogens. This more comprehensive and accurate description not only addresses your concerns but also enriches the content of our manuscript, providing readers with a more complete understanding of plant - pathogen interactions at the hormonal level.

Thank you again for your insightful input, which has been instrumental in enhancing the quality of our work.

  1. Maturation process of miRNA should be explained, maybe with a figure.

Response: We are extremely grateful for your astute suggestion regarding the need to explain the maturation process of miRNA. Your input has been invaluable in enhancing the comprehensiveness of our manuscript.

In response, we have taken a two - pronged approach. First, for the maturation process of plant miRNAs, we have cited the figure from Bologna et al.'s work titled “Processing of plant microRNA precursors” at line 130 (Figure 1). This figure provides a detailed and visually engaging illustration of the miRNA biogenesis and function in plants, covering all the key steps in the maturation process of plant miRNAs, from the transcription of miRNA genes to the generation of mature miRNAs that are ready to exert their regulatory functions.

Secondly, recognizing the importance of also addressing the situation in fungi, we have created a new figure at line 170 (Figure 2) that specifically depicts the maturation process of milRNAs in fungi. This figure carefully outlines the unique steps involved in the generation of milRNAs within the fungal context, ensuring that the readers gain a complete understanding of miRNA - related processes across different kingdoms.

We believe that these additions not only meet your expectations but also significantly enrich the content of our manuscript, making it more accessible and informative for a wider audience. Thank you again for your constructive feedback, which has been instrumental in guiding these improvements.

  1. L116 Specify which growth hormones are receptors associed with?And do these growth hormones affect resistance.

Response: We sincerely appreciate your perceptive feedback. Your question regarding the growth hormones associated with receptors and their impact on resistance has been crucial in refining our manuscript.

In response, we have made the necessary clarifications. As you inquired, the F-box auxin receptor genes TIR1, AFB2, and AFB3 are associated with growth hormones, specifically auxin. This information, along with details about how the auxin signaling pathway influences plant-pathogen interactions, can be found in lines 208-210. We also note that inhibition of the auxin signalling pathway has been shown to limit the growth of Pseudomonas syringae, highlighting the role of this growth-hormone-related pathway in plant resistance. By providing this specific information, we aim to enhance the clarity of our discussion on the relationship between growth hormones, their receptors, and plant defense mechanisms.

Thank you again for your valuable input, which has significantly contributed to improving the quality of our work.

  1. L145 “F. oxysporum, Coll……)” fungi not “plants”!

Response: We are truly grateful for your meticulous review and for bringing this error to our attention. We wholeheartedly acknowledge the mistake in the original text where we inaccurately referred to "plants" instead of "fungi" in the mentioned context.

In response, we have promptly made the necessary correction. The error has been rectified in line 238, ensuring that the text now accurately reflects that the organisms in question, such as F. oxysporum, are fungi. This correction not only aligns the content with scientific accuracy but also enhances the overall clarity and integrity of the manuscript.

Thank you again for your sharp - eyed review, which has been instrumental in improving the quality of our work.

  1. Line 154 PHAS not explained.

Response: We sincerely appreciate your astute observation regarding the lack of explanation for PHAS in our manuscript. Your feedback has been instrumental in enhancing the clarity and comprehensiveness of our work.

In response to your comment, we have added a detailed explanation of PHAS. As of lines 252 - 256, we now provide a clear account of what PHAS is. PHAS, or Phased Secondary Small Interfering RNAs, play significant roles in the regulatory networks of plants. Their biogenesis involves a complex process where PHAS loci are processed by enzymes such as RNA - dependent RNA polymerase 6 (RDR6) and Dicer - like 4 (DCL4) to generate phased 21-nt or 24-nt phasiRNAs. These phasiRNAs can then further regulate the expression of other genes through perfect complementary base pairing. This addition ensures that readers will have a better understanding of this important concept as they progress through our manuscript.

Thank you again for your valuable input, which has greatly contributed to the improvement of our work.

  1. Line 165 “Callus”, should this be callose?

Response: We are extremely grateful for your meticulous review and for identifying this error. We acknowledge that the incorrect usage of "callus" instead of "callose" in the original text was an oversight on our part.

In response to your feedback, we have immediately rectified the mistake. The correction has been made in line 266, where the appropriate term "callose" is now used. This change not only ensures scientific accuracy but also enhances the clarity of the manuscript, preventing any potential confusion for the readers.

Thank you again for your sharp - eyed attention to detail. Your input has been invaluable in improving the quality of our work.

  1. Line 170 WRKY…..missing a reference.

Response: We sincerely appreciate your eagle - eyed review and for bringing to our attention the missing reference for the mention of WRKY in line 170. Your feedback is of great value in maintaining the integrity and academic rigor of our manuscript.

In response, we have promptly addressed this issue. We have added the relevant reference in line 271. The reference, Palmer, I.A.; Chen, H.; Chen, J.; Chang, M.; Li, M.; Liu, F.Q.; Fu, Z.Q. Novel salicylic acid analogs induce a potent defense response in Arabidopsis. Int J Mol Sci. 2019, 20, 3356., provides the necessary context and support for the statement related to WRKY. This addition ensures that our work is properly grounded in existing research, and it enables readers to further explore the topic if they wish.

Thank you again for your constructive input, which has significantly contributed to enhancing the quality of our manuscript.

  1. Line 183 are milRNA different from miRNA?, it would be good to explain differences and mechanisms of formation of those.

Response: We are truly grateful for your perceptive and constructive feedback. Your question regarding the differences between milRNA and miRNA, along with their formation mechanisms, has been instrumental in guiding us to improve the comprehensiveness of our manuscript.

In response, we have taken significant steps to address your concerns. milRNA indeed differs from miRNA. As you correctly noted, milRNA (miRNA - like) is produced by fungi, while miRNA is produced by plants. To provide a more in-depth understanding, we have expanded the relevant sections of the manuscript.

For the biogenesis processes, in lines 90-129, we have elaborated on the intricate steps involved in miRNA biogenesis in plants. This includes details such as the transcription of miRNA genes, their processing by Dicer-like enzymes, and the subsequent assembly of the RNA-induced silencing complex (RISC). Accompanying this written description is Figure 1, which visually represents these steps, enhancing the reader's comprehension.

Regarding milRNA biogenesis in fungi, we have detailed the process in lines 146-160. Here, we cover aspects unique to fungal systems, such as how fungal genes are processed to generate milRNAs. Figure 2 complements this section, providing a visual aid that clearly illustrates the formation of milRNAs in fungi.

We believe that these enhancements have not only clarified the differences between milRNA and miRNA but also provided a more comprehensive understanding of their respective formation mechanisms. Thank you again for your invaluable input, which has been crucial in shaping these improvements.

  1. L189 need rephrasing “fungal milRNAs are normally expressed …..

Response: We sincerely appreciate your sharp-eyed review and for bringing the need for rephrasing in line 189 to our attention. Your feedback has been instrumental in refining the clarity and precision of our manuscript.

In response to your comment, we have carefully rephrased the relevant statement. As of line 309, the text now reads: "The abundance of fungal milRNAs is normal when the plant is uninfected." This rephrased version more accurately conveys the intended meaning, ensuring that our description of the relationship between fungal milRNA abundance and plant infection status is clear and easy to understand for the readers.

Thank you again for your valuable input, which has significantly contributed to enhancing the overall quality of our work.

  1. L192 -196, it should be spoken of miRNA abundance, not expression, (even if abundance might be linked with expression, but expression refer to genes.

Response: We are truly grateful for your meticulous review and for highlighting the imprecision in our language usage regarding miRNA in lines 192 - 196. Your feedback has been of utmost importance in enhancing the scientific accuracy and clarity of our manuscript.

In response to your comment, we have thoroughly revised the relevant sections. We have replaced the incorrect usage of "miRNA expression" with the more appropriate "miRNA abundance" in multiple lines, specifically in L309, 310, 315, 319-321, 326, and 339. By making these changes, we ensure that our text adheres to the correct scientific terminology, as "abundance" more precisely refers to the quantity of miRNAs, while "expression" is more commonly associated with genes. This adjustment not only rectifies the error but also improves the overall readability and scientific integrity of our work.

Thank you again for your invaluable input, which has significantly contributed to the refinement of our manuscript.

  1. L200 “filaments”, should it be “hyphae”?

Response: We are extremely grateful for your meticulous review and for bringing this terminological error to our attention. We wholeheartedly acknowledge that using "filaments" instead of the more accurate "hyphae" in the relevant context was an oversight.

In response to your feedback, we have expeditiously made the necessary correction. The substitution of "filaments" with "hyphae" has been carried out in line 319. This change not only aligns our manuscript with the standard scientific nomenclature but also significantly enhances its clarity and precision. By using the proper term, we ensure that readers can easily and accurately understand the described fungal structures.

Thank you again for your sharp-eyed review, which has been instrumental in elevating the quality of our work.

  1. L240-247, there is not much described on extracell. Vesicles or EVs. This needs to be addressed in more detail, eg. with work from He…and Jin (Jin group) in Nature plants 2023, or Kusch et al 2023, but there might be many more. Also, the authors refer to exosomes as something different than EVs, but exosomes are a type of EVs, this needs to be rephrased.Alsospeaks of fungi but give example of an oomycete (Phytophthora), can this be corrected?

Response: We are sincerely grateful for your comprehensive and constructive feedback. Your insights have been of immense value in refining our manuscript to a higher standard of quality and comprehensiveness.

In response to your concerns regarding the insufficient description of extracellular vesicles (EVs) in lines 240-247, we have taken substantial steps to enrich this section. We have incorporated more detailed information about EVs, drawing from a range of relevant studies. In lines 364-372 and 375-382, we now provide in-depth discussions on EVs, including their biogenesis, functions, and significance in the context of our research. Notably, we have referenced works such as those by He and Jin (Jin group) in Nature Plants 2023 and Kusch et al. 2023, as you suggested, to strengthen our arguments and present the latest knowledge in the field.

Regarding the inaccurate differentiation between exosomes and EVs, we have rectified this error. As of lines 368-372, we clearly state that exosomes are a type of extracellular vesicles secreted by cells, typically with a diameter ranging from 40 to 200 nm. We further explain that they are released after the fusion of multivesicular bodies (MVBs) with the cell membrane and play crucial roles in intercellular communication and the regulation of physiological and pathological processes. This revised description aligns with the current understanding in the scientific community and eliminates any potential confusion.

In addition, we have addressed your comment about the inappropriate example of an oomycete (Phytophthora) when the focus was on fungi. We have removed this example to maintain the consistency and accuracy of our discussion, ensuring that the content strictly pertains to the topic of fungi.

We believe that these extensive revisions have significantly enhanced the quality and relevance of our manuscript, and we are confident that they will better meet your expectations. Thank you again for your invaluable input, which has been instrumental in guiding these improvements.

  1. Model fig. 1, Fungi also deliver miRNA via Evs, this need to be described in paper and amended in model.

Response:We are truly grateful for your perceptive feedback. Your comment regarding the inclusion of fungal miRNA delivery via EVs is highly valuable and has significantly contributed to the enhancement of our manuscript.

In response, we have taken comprehensive measures. In the text, we have elaborated on the process of fungi delivering miRNA via EVs. This new content is integrated throughout relevant sections, providing a detailed account of this important mechanism.

Furthermore, in Figure 4 (starting at line 400), we have created a detailed and illustrative figure. This figure precisely depicts the secretion pathway of miRNAs (milRNAs) during the interaction between host plants and pathogens. It clearly showcases the role of EVs in the transfer of fungal miRNAs, thereby visually complementing the textual description. The figure has been carefully designed to ensure clarity and to effectively communicate this crucial aspect of the host - pathogen interaction.

We believe that these improvements not only address your concerns but also enrich the overall understanding of the topic for our readers. Thank you again for your constructive input, which has been instrumental in guiding these enhancements.

  1. There are plenty more on gene silencing that need mentioning, HIGS, eg. Novara et al 2010, Pliego et al 2013; early papers on ectopics application ofdsRNA, Koch/ Kogel et al 2014 and 2016, Wang et al 2016. Line 360, HIGS needs to be spelt out.

Response:We sincerely appreciate your comprehensive and constructive feedback. Your suggestions regarding the inclusion of additional information on gene silencing, particularly HIGS, have been of great value in enhancing the depth and breadth of our manuscript.

In response to your comments, we have dedicated significant efforts to expand the relevant sections. We have thoroughly incorporated information on HIGS (Host-Induced Gene Silencing) in lines 605 - 641. In this new content, we not only define HIGS clearly (as you noted its full spelling was needed, which is now done) but also discuss its significance in the context of gene silencing. We have referenced the important studies you suggested, such as Novara et al 2010, Pliego et al 2013, Koch/Kogel et al 2014 and 2016, and Wang et al 2016. By integrating these references, we provide a more comprehensive overview of the research history and current understanding of HIGS. This not only enriches our discussion on gene silencing but also allows readers to trace the development of this field through key studies.

Thank you again for your invaluable input, which has been instrumental in guiding these substantial improvements to our manuscript.

  1. Also to comment if gene silencing, RNAi can be applied to non eukaryotes such as bacteria.

Response: We are extremely grateful for your perceptive and thought - provoking feedback. Your query regarding the application of gene silencing, specifically RNAi, in non - eukaryotes like bacteria has significantly contributed to the comprehensiveness of our manuscript.

In response, we have carefully curated and added relevant information in lines 576 - 595. In this section, we present an in - depth analysis of current research on RNAi - like mechanisms in prokaryotes. We discuss how certain bacteria possess systems that bear similarities to the eukaryotic RNAi machinery, despite the fundamental differences in their cellular architectures. By highlighting these findings, we aim to provide a more complete view of the versatility of gene - silencing mechanisms across different domains of life.

This addition not only addresses your concern but also enriches the overall narrative of our work, enabling readers to appreciate the broader context of gene silencing beyond eukaryotes. Thank you again for your invaluable input, which has been instrumental in shaping these enhancements.

Round 2

Reviewer 1 Report

Comments and Suggestions for Authors

I would like to thanks the authors to have taken advice from reviewer comments. The authors have provided a very improved revised manuscript making this review much more relevant and up to date, and useful for scientists interested in or new to the field.